# OpenReview forum: "Emergence of Hierarchical Emotion Representations in Large Language Models"
_ICLR.cc/2025/Conference — Submitted to ICLR 2025_

### Official Review · Reviewer_6r52 · 2024-10-30

**Soundness:** 4
**Presentation:** 3
**Contribution:** 4
**Rating:** 6
**Confidence:** 3

**Summary:**

This study reveals three findings about emotional intelligence in large language models (LLMs) and its practical implications. First, as LLMs scale up, they develop hierarchical representations of emotions that align with psychological models. Second, the study uncovers how different personas (based on gender, socioeconomic status, etc.) can bias LLMs' emotion recognition, particularly showing systematic biases for minority attributes. Finally, through a synthetic sales negotiation task, the research demonstrates that better emotional prediction capabilities directly correlate with improved persuasion and negotiation outcomes.

**Strengths:**

There are three main strengths of this paper:

1. **High originality**: The exploration of hierarchical emotional representations in LLMs is novel and important. While previous studies have examined emotions in LLMs from various angles, none have investigated their hierarchical nature. Additionally, few works have explored the personalization of emotions, which this paper thoroughly investigates.


2. **High quality and clarity**: The paper presents solid evidence through multiple experiments and maintains clear, fluid expression throughout.


3. **Significant impact**: The findings on emotional hierarchy and personalized emotional biases provide valuable insights for future research and have important implications for LLMs' emotional reasoning and recognition.

**Weaknesses:**

There are three main weaknesses of this paper:

1. **Limited data for emotion tree construction**: The study utilized 5,000 prompts to test 135 emotion types, resulting in an average of only 37.03 prompts per emotion type. This relatively small sample size per emotion suggests the need for expanded data collection to construct a more detailed and robust emotion tree.

2. **Dataset limitations**: The study exclusively relies on GPT-4 generated datasets. It would benefit from incorporating data from real-world scenarios (such as EDOS[1], EmpatheticDialogues[2], and GoEmotions[3]) for experimental validation.

3. **Format error**: (3.1) Citation formats require standardization (inconsistencies noted in lines 34, 48, 249, and 250); (3.2) Possible typo: "eutral" appears on line 362 (should this be "neutral"?)

[1] A taxonomy of empathetic response intents in human social conversations

[2] Towards empathetic opendomain conversation models: A new benchmark and dataset.

[3] Goemotions: A dataset of fine-grained emotions.

**Questions:**

1. **Methodology Consideration**: Did the research employ various prompt types when constructing the emotion tree? Given that LLMs are typically sensitive to prompting, different prompt structures might elicit varying responses. This could potentially affect the emotional relationships identified in the study - for example, the strong connection observed between fear and shock in Llama3.1_8b might be weakened or altered with different prompt formulations. Therefore, I suggest conducting an ablation study on prompt sensitivity to quantify how different prompts affect the emotional hierarchy.

2. **Data Representation Query**: Considering that all the data used in the study was generated by GPT-4o, to what extent might this deviate from authentic human emotional expressions and patterns? I recommend that the authors compare GPT-4o generated data with existing human-annotated emotion datasets to quantify any differences. Additionally, human experts could evaluate the differences between GPT-4o generated data and real-world data.

---

> ### Author Response · Authors · 2024-11-23
> **Response to Reviewer Comments**
>
> Thank you for recognizing the originality, quality, and impact of our paper. We also appreciate your suggestions on how to improve this paper, which we address below.
>
> ## Regarding Weaknesses
>
> > 1. Limited data for emotion tree construction: The study utilized 5,000 prompts to test 135 emotion types, resulting in an average of only 37.03 prompts per emotion type. This relatively small sample size per emotion suggests the need for expanded data collection to construct a more detailed and robust emotion tree.
>
> We agree that expanding the dataset could enhance the robustness and granularity of the constructed emotion tree. While our current dataset provides a proof-of-concept, we acknowledge that a larger sample size per emotion type would help refine the hierarchical structure and capture nuances more effectively. As part of future work, we plan to scale up the number of prompts to provide greater coverage for each emotion type. Additionally, we hope to explore methods for leveraging external datasets or employing data augmentation techniques to further strengthen the emotion tree construction.
>
> > 2. Dataset limitations: The study exclusively relies on GPT-4 generated datasets. It would benefit from incorporating data from real-world scenarios (such as EDOS[1], EmpatheticDialogues[2], and GoEmotions[3]) for experimental validation.
>
> Thank you for your insightful suggestion. We conducted an additional experiment using the GoEmotions dataset for you. The results have been added to Figure 10 and are discussed in the paragraph titled "Expanding to realistic datasets" in Section 4.1. This experiment further supports our findings by highlighting biases in emotion recognition observed in both humans and LLMs, thereby strengthening our conclusions. We greatly appreciate your feedback!
>
> > 3. Format error: (3.1) Citation formats require standardization (inconsistencies noted in lines 34, 48, 249, and 250); (3.2) Possible typo: "eutral" appears on line 362 (should this be "neutral"?)
>
> Thank you for pointing these out. We have fixed the citation formats and typos in the updated draft.
>
>
> ## Response to questions
>
> > 1. Methodology Consideration: Did the research employ various prompt types when constructing the emotion tree? Given that LLMs are typically sensitive to prompting, different prompt structures might elicit varying responses. This could potentially affect the emotional relationships identified in the study - for example, the strong connection observed between fear and shock in Llama3.1_8b might be weakened or altered with different prompt formulations. Therefore, I suggest conducting an ablation study on prompt sensitivity to quantify how different prompts affect the emotional hierarchy.
>
> Due to resource constraints, we are not able to rerun the hierarchy experiments before the end of the rebuttal period. However, this is a great suggestion, and we will add an ablation study to the final version of the paper.
>
> > 2. Data Representation Query: Considering that all the data used in the study was generated by GPT-4o, to what extent might this deviate from authentic human emotional expressions and patterns? I recommend that the authors compare GPT-4o generated data with existing human-annotated emotion datasets to quantify any differences. Additionally, human experts could evaluate the differences between GPT-4o generated data and real-world data.
>
> Thank you for your thoughtful suggestion! In response to your comment, we have:
> - Conducted an experiment using the human-annotated emotion dataset, GoEmotions [3], to quantify differences between human data and LLM-generated data (Figure 10).
> - Performed a user study with 60 participants to evaluate differences in emotion recognition between humans and LLMs (Figure 9, 20 and 22).
> - Added new experiments with a stronger focus on psychological perspectives, connecting our findings to psychology literature (Figures 11 and 12).
>
> We've added a new Section 4.1 to provide a detailed discussion of these results. Interestingly, LLM shows strong overall emotion recognition abilities but can also display human-like behavior when adopting specific personas, as highlighted in our discussion. Additionally, we've clarified how our experiments are grounded in psychology research by improving the introduction (Section 1) and refining the opening of Section 4. We appreciate your thoughtful feedback and are happy to address any additional suggestions you may have!

---

> ### Comment · Reviewer_6r52 · 2024-11-26
> **Response to Authors**
>
> Thank you for your response; you have addressed my concerns.
>
> Overall, I believe this paper has value and can help future work clarify the emotional mechanisms within LLMs. The research content is quite comprehensive and convincing.
>
> However, I think the paper has some minor flaws. The most critical part is the construction of the hierarchical emotion tree. This section's construction strategy relies solely on statistical metrics and lacks significant innovation, despite the interesting motivation.
>
> In summary, **I believe the accurate rating should be 6.5**. However, the system does not offer this option, so I have not adjusted the score.

---

### Official Review · Reviewer_9Uj3 · 2024-11-04

**Soundness:** 2
**Presentation:** 3
**Contribution:** 3
**Rating:** 3
**Confidence:** 4

**Summary:**

This paper studies the emergence of hierarchical emotional representations in large language models and explores their abilities to predict and manipulate emotions. The focus of this study is on models such as LLaMA and GPT, analyzing their emotional hierarchies and the potential biases they may exhibit when identifying emotions of minority character roles. The study also assesses the performance of models in comprehensive negotiation tasks, revealing the correlation between emotional prediction accuracy and negotiation outcomes.

**Strengths:**

•  The analysis and extraction of the emotional hierarchy in LLaMA validate its similarity to human emotional structures, with the complexity of emotional hierarchy positively correlated with model parameter volume.

•  It validates that different roles and scenarios significantly affect LLMs’ emotion recognition abilities, providing guidance for how to avoid such biases in the future.

•  It analyzes the connection between emotional prediction ability and persuasive ability in negotiation tasks, offering practical insights for the application of artificial intelligence in emotionally sensitive environments.

**Weaknesses:**

•  The first two conclusions are quite obvious and lack in-depth exploration of their underlying causes. For example, what is the relationship between the breadth and depth of model emotional stratification and model parameters and pre-training corpora?

•  The discussion on ethics and biases is somewhat coarse in terms of categorization by region, ethnicity, cultural background, and other living conditions.

•  There is a lack of discussion on how to leverage LLMs’ emotional prediction capabilities to optimize downstream dialogue tasks.

**Questions:**

•  The bias experiment could be expanded to more detailed demographic attributes or a broader set of test roles.

•  The analysis of the relationship between emotional prediction and other abilities (such as negotiation, persuasion) could be further expanded, rather than being limited to sales.

•  The wording around ethical issues in the abstract and introduction could be strengthened by providing specific examples of potential real-world impacts.

•  The presentation of Fig 6 needs to be optimized, with biases of different roles not being prominent enough.

---

> ### Author Response · Authors · 2024-11-23
> **Response to Reviewer Comments [1/2]**
>
> Thank you for your thoughtful feedback. We’re glad you appreciated our method for studying hierarchical emotion representations in LLMs, measuring bias and persuasion abilities, and our practical insights for applying AI in emotionally sensitive environments. We address your suggestions below.
>
> ## Regarding Weaknesses
> > 1. The first two conclusions are quite obvious and lack in-depth exploration of their underlying causes. For example, what is the relationship between the breadth and depth of model emotional stratification and model parameters and pre-training corpora?
>
> We share the reviewer’s interest in how particular factors such as model size and pre-training data influence emotion representations. Indeed, one of our goals in Section 3 was to explore how different size LLMs represent emotions.  As noted by the reviewer, our analysis reveals two critical dimensions of hierarchical emotion representations that vary across model size: breadth (related to node count) and depth (measured through hierarchical relationships). More specifically we find that while both dimensions generally increase with model size, emotional breadth appears to grow with parameter count, while depth shows diminishing returns beyond certain model sizes.
>
> This relationship is demonstrated in Figure 3, where we compare models with 8B, 70B, and 405B parameters. While the 8B model shows only basic emotion groupings, the 70B model demonstrates increased sophistication with four layers. The 405B model shows the most nuanced understanding with a five-layer structure, correctly placing embarrassment and humiliation as subcategories of shame, while grouping thrill and enthusiasm under excitement—arrangements that align well with human psychological understanding.
>
> While we did not explore the impact of pre-training data on emotion representations in this work, we see this as a promising avenue for future work. In particular, it may be that better understanding which data leads to which emotion representations may help us better fine-tune models to improve the complexity of their emotional understanding, or to mitigate harmful biases.
>
> > The discussion on ethics and biases is somewhat coarse in terms of categorization by region, ethnicity, cultural background, and other living conditions.
>
> Thank you for your thoughtful suggestion. Following your comments as well as similar points raised by other reviewers,  we have added 20+ new experiments to our paper, including:
> - Detailed demographics: Groups categorized by income (high/low), race (Black/White), and gender (man/woman), creating 2x2x2 combinations.
> - Broader test roles: Expanded categories such as race (e.g., White, Black, Hispanic, Asian), religion (e.g., Christian, Muslim, Buddhist, Hindu), and psychological conditions (e.g., ASD, depression, anxiety). Including psychological conditions allows our findings to connect with existing psychological research.
> - Granular roles: A breakdown of education levels (e.g., postgraduate, college graduate, some college, high school, less than high school).
>
> These new findings are included in Figure 6, 7, 8, 20, 21 and 22 with detailed discussions in Section 4. Additionally, we plan to analyze data by country to explore cultural dimensions and refine demographics for deeper intersectionality insights. We would be very happy to incorporate any further suggestions you may have.
>
> > 3. There is a lack of discussion on how to leverage LLMs' emotional prediction capabilities to optimize downstream dialogue tasks.
>
> Thank you for your suggestion. The emotion tree provides distance measures to quantify the depth of LLMs' understanding of the relationships between emotions. By leveraging such distance measures within the emotion tree as a reward for the model, we expect it to achieve better performance in downstream tasks such as persuasion and negotiation. We have added this discussion to Section 3.2.

---

> > ### Author Response · Authors · 2024-11-23
> > **Response to Reviewer Comments [2/2]**
> >
> > ## Response to questions
> > > The bias experiment could be expanded to more detailed demographic attributes or a broader set of test roles.
> >
> > See our response to Weakness 2.
> >
> > > The analysis of the relationship between emotional prediction and other abilities (such as negotiation, persuasion) could be further expanded, rather than being limited to sales.
> >
> > Thank you for your excellent feedback! Inspired by your comment, we conducted a new experiment beyond the sales scenario, focusing on handling customer complaints. The results are now included in Figure 13(b) in Section 5. In this experiment, we evaluated the ability to manipulate emotions by measuring the degree of anger reduction in the customer complaint scenario. We found that improved emotion prediction can enhance the effectiveness of emotion manipulation in this additional scenario as well.
> >
> > > The wording around ethical issues in the abstract and introduction could be strengthened by providing specific examples of potential real-world impacts.
> >
> > We have improved the presentation of our paper to highlight the potential real-world impacts, by adding citations and example areas of ethical issues. For instance, we explore how biases in emotion recognition capabilities of LLMs could be exploited, particularly in scenarios where AI agents might deceptively manipulate user emotions while evading human oversight. To further demonstrate LLMs' capabilities in realistic contexts, we have expanded Section 5 to include our original sales simulation, now enhanced with granular persona adaptation, as well as a new compliance simulation study.
> >
> > > The presentation of Fig 6 needs to be optimized, with biases of different roles not being prominent enough.
> >
> > Thank you for your valuable suggestion. To enhance clarity, we have updated Figure 6 to include a bar plot that more effectively highlights the differences between roles. Additionally, we conducted new experiments incorporating detailed demographic attributes, now presented in Figure 6(g). These updates allow for an analysis of intersectional bias and provide further evidence supporting our finding that LLMs exhibit bias against minority groups in emotion recognition.
> > Furthermore, thanks to your comment, we have now prepared Java-script based interactive project website, included as supplementary material (Please see the general response for access to the demo).

---

> > > ### Author Response · Authors · 2024-11-30
> > > **Summary of updates before discussion deadline**
> > >
> > > Dear Reviewer 9Uj3,
> > >
> > > We sincerely thank you for your thoughtful feedback that helped us strengthen the practical and ethical dimensions of our work. As the discussion period ends in two days, we would like to **summarize how we've addressed your specific concerns**:
> > >
> > > 1. **Expanded demographic analysis:**
> > >    * Conducted **new user study with 60 human participants** to validate emotion recognition patterns (**new Fig. 9**)
> > >    * Added comprehensive analysis across detailed demographic attributes (**new Fig. 6b-g**)
> > >      - Race (White, Black, Hispanic, Asian)
> > >      - Religion (Christian, Muslim, Buddhist, Hindu)
> > >      - Education levels (postgraduate through high school)
> > >      - Intersectional analysis with 2x2x2 combinations of income/race/gender
> > >    * Enhanced visualization of demographic-specific biases (**new Fig. 7, 8**)
> > >    * Developed interactive visualization tool to explore detailed results (available in supplementary materials)
> > >
> > > 2. **Beyond sales scenarios:**
> > >    * Added **new customer complaint handling scenario** (**new Fig. 13b**)
> > >    * Demonstrated correlation between emotion prediction accuracy and anger reduction
> > >    * Added granular persona adaptation analysis with different demographics (**new Fig. 25**)
> > >
> > > 3. **Real-world impacts:**
> > >    * Strengthened ethical analysis with concrete examples
> > >    * Added validation through comparison with human-annotated emotion hierarchies (**new Fig. 16, 17**)
> > >    * Added analysis of how emotion prediction capabilities affect manipulation outcomes (**new Fig. 24**)
> > >    * Added new experiments showing bias amplification in intersectional underrepresented groups (**new Fig. 8**)
> > >
> > > Given these substantial updates expanding both our demographic analysis and real-world implications, we would greatly appreciate your feedback on whether these changes adequately address your concerns. If you have any remaining questions or need clarification, we are ready to respond promptly.

---

### Official Review · Reviewer_qTjt · 2024-11-04

**Soundness:** 2
**Presentation:** 2
**Contribution:** 2
**Rating:** 5
**Confidence:** 2

**Summary:**

This paper explores the development of hierarchical emotion representations in large language models (LLMs), particularly focusing on models like LLaMA 3.1 with up to 405B parameters. The authors propose methods to extract emotion hierarchies from LLM outputs by analyzing probabilistic dependencies between emotional states. They claim that larger models exhibit more intricate emotional hierarchies resembling psychological theories of emotion. Additionally, the paper examines the impact of persona biases (e.g., gender, socioeconomic status) on emotion recognition and explores the relationship between emotional modeling and persuasive abilities in synthetic negotiation tasks.

**Strengths:**

1. Innovative Approach: The paper introduces a novel and interesting methodology for extracting hierarchical structures of emotions from LLMs, bridging computational models with psychological frameworks.

2. Relevance and Timely: The topic is timely, addressing the intersection of AI, emotion modeling, and ethics.

**Weaknesses:**

1. Emotion Extraction Technique Concern: The method for extracting hierarchical structures based on next-word probabilities lacks rigorous justification. There is no comparison with alternative methods or validation.

2. Threshold Selection: The paper sets a threshold (0 < t < 1) for determining parent-child relationships but does not explain how this threshold is chosen or its impact on the results.

3. Quantitative Metrics: Although the visual representations of emotion hierarchies are compelling, incorporating additional quantitative metrics or comparisons with human-annotated emotion hierarchies could provide stronger validation of the proposed method.

4. The font in Figure 2 is too small to see easily.

**Questions:**

1.  Can you provide a more detailed justification for using next-word probabilities to extract hierarchical emotion structures?

2. How did you determine the appropriate threshold value (0 < t < 1) for establishing parent-child relationships between emotions? Was this threshold empirically validated?

3. Besides visual representations, can you use some quantitative metrics to validate the integrity and accuracy of the extracted hierarchical emotion structures?

4. Besides emotion, I guess your method can visualize the structure of other entities. Can you extend this part more to enlarge the generalization of your method?

---

> ### Author Response · Authors · 2024-11-23
> **Response to Reviewer Comments [1/2]**
>
> Thank you for the detailed comments and for appreciating the novelty and timeliness of our work. We appreciate your suggestions, which we address below.
>
> ## Weakness 1: Emotion Extraction Technique
> > Emotion Extraction Technique Concern: The method for extracting hierarchical structures based on next-word probabilities lacks rigorous justification. There is no comparison with alternative methods or validation.
> ### Question 1
> > Can you provide a more detailed justification for using next-word probabilities to extract hierarchical emotion structures?
>
> Thank you for this question about our methodological approach. We have added a detailed explanation of our probabilistic framework in Appendix A. Our prompting format ("[Emotion scenario.] As a [person from a specific demographic group], I think the emotion involved in this situation is") allows us to interpret next-word probabilities as approximations of the model's estimated likelihood for each emotion given the scenario. Building on this interpretation, Appendix A provides a formal probabilistic foundation for our hierarchical structures. The elements in our matching matrix represent the joint probabilities of emotion pairs co-occurring across scenarios. Furthermore, our criteria for including edges in the hierarchy are based on conditional probabilities between emotions—specifically, how strongly the presence of one emotion predicts another. This probabilistic framework provides theoretical grounding for our extraction of hierarchical emotion structures from language model predictions.
>
> ## Weakness 2: Threshold Selection
> > Threshold Selection: The paper sets a threshold (0 < t < 1) for determining parent-child relationships but does not explain how this threshold is chosen or its impact on the results.
> ###  Question 2
> > How did you determine the appropriate threshold value (0 < t < 1) for establishing parent-child relationships between emotions? Was this threshold empirically validated?
>
> Thank you for your valuable feedback. We sweeped threshold and selected the threshold to ensure the hierarchy trees have a reasonable number of edges and nodes, with the largest tree's number of nodes approximately equal to the number of emotions considered (135). We have now included plots illustrating the threshold sweep (see Figure 15 in Appendix D). We can see that the increasing trend with model size remains robust across different threshold selections.
>
> ## Weakness 3: Quantitative Metrics
> > Quantitative Metrics: Although the visual representations of emotion hierarchies are compelling, incorporating additional quantitative metrics or comparisons with human-annotated emotion hierarchies could provide stronger validation of the proposed method.
> ### Question 3
> > Besides visual representations, can you use some quantitative metrics to validate the integrity and accuracy of the extracted hierarchical emotion structures?
>
> Thank you for your insightful feedback. Based on your suggestions, we included a quantitative comparison between the extracted hierarchical emotion structures (Figure 3) and the human-annotated emotion hierarchies (i.e., the emotion wheel shown in Figure 1) in Figure 16. We observed significant correlations (correlation > 0.5, p < 0.001) between the node distances in the hierarchical tree and their corresponding distances on the emotion wheel across the LLMs, supporting the accuracy of the LLM-derived emotion structures.
>
> We also want to emphasize that traditional emotion wheels have been largely intuition-driven, as their empirical construction through human experiments faces significant challenges. These include the need for a large number of participants, the overwhelming difficulty for humans to evaluate relationships among all pairs of 135 emotion classes, and the influence of subjective biases. By leveraging the vast corpora accessible to LLMs, we demonstrate that emotion wheels can now be constructed in a fully data-driven manner. The reconstructed hierarchical trees provide a more granular and nuanced representation of emotional relationships compared to the simpler two-layer structure of traditional emotion wheels. Our work, originally inspired by psychology and cognitive science, might now have the potential to offer new insights back to these fields.

---

> ### Author Response · Authors · 2024-11-23
> **Response to Reviewer Comments [2/2]**
>
> ## Weakness 4: Figure Readability
> > The font in Figure 2 is too small to see easily.
>
> We have increased the font size in Figure 2 (now Figure 3) to improve readability. Additionally, in response to your feedback, we have developed a JavaScript-based interactive project website, included as supplementary material (see the general response for access to the demo). This platform allows users to dynamically adjust scaling, enhancing both accessibility and interactivity.
>
> ### Question 4
> > Besides emotion, I guess your method can visualize the structure of other entities. Can you extend this part more to enlarge the generalization of your method?
>
> Thank you for your insightful suggestion! Inspired by your comment, we conducted an additional experiment in another domain: scent. We identified a reasonable hierarchical relationship between aroma words, aligning with human-annotated aroma wheel. The results are now included in Figure 19, demonstrating the generality of our method.

---

> ### Author Response · Authors · 2024-11-30
> **Summary of updates before discussion deadline**
>
> Dear Reviewer qTjt,
>
> We sincerely thank you for your thoughtful feedback that highlighted critical methodological aspects requiring stronger validation. As the discussion period ends in two days, we would like to summarize how we've addressed your key concerns:
>
> 1. **Justification of emotion extraction method:**
>    * Added formal probabilistic framework and mathematical foundations in Appendix A
>    * Added validation of our method by comparing results with human-annotated emotion hierarchies (**new Fig. 16** and **17**)
>    * Demonstrated strong correlation (>0.5, p<0.001) between our extracted structures and established psychological frameworks
>    * Conducted new **user study with 60 human participants** to validate emotion recognition patterns (**new Fig. 9**)
>
> 2. **Threshold selection and validation:**
>    * Added comprehensive threshold sweep analysis (**new Fig. 15**)
>    * Demonstrated robustness of our findings across different threshold values
>    * Added quantitative criteria for threshold selection, targeting approximately 135 nodes to match the number of emotions considered
>
> 3. **Quantitative metrics and method generalization:**
>    * Added correlation analysis between hierarchical tree distances and emotion wheel distances (**new Fig. 16**)
>    * Expanded methodology to a new domain by successfully applying our tree-construction algorithm to wine aroma hierarchies (**new Fig. 19**)
>    * Validated results through comparison with the established Davis Wine Aroma Wheel, demonstrating the generalizability of our approach
>
> Given these substantial updates strengthening the methodological foundations of our work, we would greatly appreciate your feedback on whether these changes adequately address your concerns. If you have any remaining questions or need clarification, we are ready to respond promptly.

---

### Official Review · Reviewer_Lh9i · 2024-11-04

**Soundness:** 3
**Presentation:** 4
**Contribution:** 2
**Rating:** 6
**Confidence:** 3

**Summary:**

This study reveals key advancements in how LLMs perceive, predict, and influence human emotions. As model size increases, LLMs develop hierarchical emotional representations consistent with psychological models. The research highlights that personas can bias emotion recognition, underscoring the risk of stereotype reinforcement. Additionally, the study demonstrates that LLMs with refined emotional understanding perform better in persuasive tasks, raising ethical concerns about potential manipulation of human behavior. These insights call for robust ethical guidelines and strategies to mitigate risks of emotional manipulation.

**Strengths:**

1. The writing in this paper is clear, and the figures are intuitive, making the author's ideas easy to understand.

2. The paper astutely identifies that LLMs' potential to comprehend emotions could enhance their capacity to manipulate emotions, which provides critical ethical considerations for the further development of LLMs.

3. The proposed hierarchical emotion extraction method appears simple and effective, offering a powerful tool for further analysis.

**Weaknesses:**

1. After reading the introduction, I expected Chapter 3 to discuss the model's ability and limitations in **perceiving** emotions, especially focusing on the circumstances under which the model fails. However, the paper mainly discusses how larger models outperform smaller ones in understanding emotions, which is rather obvious and does not provide sufficient novel insight.

2. Chapter 4 employs synthetic data for testing but lacks sufficient quality validation. Including human and LLM prediction accuracy in a figure, such as Figure 6, would be beneficial, even if only for a subset.

3. The contributions of the paper are somewhat scattered, covering three different aspects, but the discussions on these points are inadequate. Given that “influencing human emotions” is highlighted as a major contribution, I expected more extensive coverage on this topic. While I understand that involving human subjects may incur additional costs, drawing conclusions solely from LLM dialogues in isolated scenarios lacks persuasiveness. This section also lacks deeper analysis.

**Questions:**

1. What is the underlying principle behind Chapter 3? Your algorithm extracts more nuanced and hierarchical emotional information, but can you elaborate on what further conclusions can be drawn from this? If I understand correctly, does the model's ability to use more emotion-related vocabulary lead to greater hierarchical richness?

2. Chapter 4 provides quantitative analysis from multiple perspectives, but could you offer specific examples of how different character background settings lead to different model emotion predictions? This would help provide more substantial insights.

3. Could you clarify what new insights your experiments provide to advance previous work in perceiving, predicting, and potentially influencing human emotions? Some aspects have been discussed individually in previous studies.

---

> ### Author Response · Authors · 2024-11-23
> **Response to Reviewer Comments [1/3]**
>
> Thank you for your detailed comments. We are delighted that you found our hierarchical emotion extraction method effective and our paper clear. We are happy to address your comments below.
>
> ## Regarding Weaknesses
> ### 1. Model Limitations and Emotion Perception
> > After reading the introduction, I expected Chapter 3 to discuss the model's ability and limitations in perceiving emotions, especially focusing on the circumstances under which the model fails. However, the paper mainly discusses how larger models outperform smaller ones in understanding emotions, which is rather obvious and does not provide sufficient novel insight.
>
> Inspired by your comment, "discuss the model's ability and limitations in perceiving emotions, especially focusing on the circumstances under which the model fails.", we now have included a new set of "human experiments" to see where and how the models' prediction differs in emotion state prediction. We find that LLM struggles to recognize the emotion of "surprise," but demonstrates a greater overall ability in emotion perception compared to humans (Figure 9a). Interestingly, LLMs can also "fail" like humans when adopting certain personas (Figure 9b and 9c).
>
> > However, the paper mainly discusses how larger models outperform smaller ones in understanding emotions, which is rather obvious and does not provide sufficient novel insight.
>
> Thank you for your thoughtful feedback. One of our key contributions is introducing a new problem: evaluating LLMs' intrinsic understanding of emotion recognition. To address this, we developed a novel algorithm that goes beyond standard benchmark metrics. While many existing approaches tackle similar tasks, most of them overlook the hierarchical relationships among emotions, a well-established concept in psychology. Hierarchical clustering algorithms assume hierarchical relationships between clusters, but not among emotions themselves. Our work is the first to explore LLMs' intrinsic understanding of relationships among emotions purely through their internal representations. Our tree-construction approach is deeply inspired by the concept of emotion differentiation and "emotion wheel" (Figure 1), pioneered and widely used in cognitive psychology. Our algorithm can capture the hierarchical relationships between pairs of emotions and reconstruct the overall structure of the emotional hierarchy. By applying our proposed algorithm, we uncovered a nontrivial phenomenon: LLMs' intrinsic understanding of emotions emerges as the model size increases. To clarify these points further, we have updated the related work section and Section 3.2 to ground our methodology and insights more clearly.
>
> ### 2. Synthetic Data Testing and Validation
>
> > Chapter 4 employs synthetic data for testing but lacks sufficient quality validation. Including human and LLM prediction accuracy in a figure, such as Figure 6, would be beneficial, even if only for a subset.
>
> Thank you for your suggestion to include human experiments! We conducted new user experiments and generated a figure analogous to Figure 6. These new results are presented in Figure 9, and the corresponding discussions are included in the paragraph titled *"User Study: Comparing emotion recognition in humans and LLMs."* in Section 4.1. In summary, while LLMs have superior emotion recognition ability than human overall, it has human-like bias in emotion recognition accuracy and misclassification patterns when adopting specific personas. Our user study involved 60 participants due to time constraints; however, we plan to scale this up to a larger sample size for the camera-ready version to ensure robustness and generalizability of our findings.

---

> ### Author Response · Authors · 2024-11-23
> **Response to Reviewer Comments [2/3]**
>
> ### 3. Paper Structure and Contributions
> > The contributions of the paper are somewhat scattered, covering three different aspects, but the discussions on these points are inadequate.
>
> Thank you for this important observation about the paper's structure. We have made substantial revisions together with new experimental results (Section 4 and 5) to create a more cohesive narrative that follows a clear logical progression through our three main contributions.
>
> Our investigation began with a fundamental question: Do LLMs have a human-like understanding of emotions? In Section 3, using a novel tree-construction algorithm inspired by established psychological concepts, we demonstrated that LLMs do exhibit human-like representation of emotional hierarchies. This finding naturally led to two follow-up questions: Can LLMs accurately perceive human emotions adapting various personas (Section 4), and how might this capability affect their ability to influence emotions through dialogues (Section 5)?
>
> Following your feedback, we've now strengthened the connections between these sections. The detailed persona analysis newly introduced in Section 4's emotion classification study now extends into Section 5. To reflect your feedback, we now examine
> - **How LLMs' intrinsic understanding of emotions**, captured through the geometric measures of the emotion trees, **affects emotion recognition accuracy**. Our new analysis, presented in Figures 23 and 24, demonstrates that the structure of the emotion tree significantly influences recognition performance, highlighting the relationship between hierarchical understanding and accuracy. This analysis tightly links Sections 3 and 4.
> - **How biases in emotion recognition affect persuasion outcomes in sales scenarios**. Specifically, we conducted new experiments using personas with 2x2x2 combinations of education level (high/low), race (Black/White), and gender (man/woman). Our findings in Figure 25 reveal that combinations of underrepresented attributes correlate with both lower emotion predictions and decreased ability to manipulate, consistent with the biases observed in Section 4, thus tightly weaving those sections (Section 4 and 5).
>
> > While I understand that involving human subjects may incur additional costs, drawing conclusions solely from LLM dialogues in isolated scenarios lacks persuasiveness.
>
> Thank you for your thoughtful feedback. We understand the need for more comprehensive coverage on influencing human emotions. To address this, we'll expand emotion dynamics experiments (Section 5) by including dialogue tasks with human participants to validate our conclusions.
>
> ## Response to Questions
>
> ### 1. Underlying Principles and Conclusions
>
> > What is the underlying principle behind Chapter 3? Your algorithm extracts more nuanced and hierarchical emotional information, but can you elaborate on what further conclusions can be drawn from this? If I understand correctly, does the model's ability to use more emotion-related vocabulary lead to greater hierarchical richness?
>
> The principle and goal of Chapter 3 is to extract the "structure of emotion representation" in LLMs, going beyond traditional benchmarking metrics like classification accuracy. Our hierarchical analysis provides several key insights:
>
> First, the extracted tree structure reveals two important dimensions:
> - The breadth of emotional understanding (represented by the number of nodes)
> - The depth of emotional comprehension (shown through hierarchical relationships)
>
> The number of nodes correlates with the LLM's vocabulary size of emotions, while tree depth indicates how sophisticated the model is in grouping related emotions. Thanks to your comments, we included this discussion in Section 3.2.
>
> This is clearly demonstrated in Figure 3, where larger LLM models show increased complexity in both dimensions. For instance, the Llama-405B model demonstrates particularly nuanced understanding through its five-layer hierarchical groupings. It correctly places embarrassment and humiliation as subcategories of shame, while clustering thrill and enthusiasm under excitement – arrangements that align well with human psychological understanding.
>
> Our methodology also offers novel insights into emotional understanding across different demographic groups and psychological conditions. As shown in Figure 12, by adapting personas (such as individuals with ASD), we can analyze how emotional representation structures vary across different populations. While traditional psychology has explored these relationships through careful methods involving human studies, our approach offers a unique, data-driven technique for rapidly extracting these structures. This represents a novel approach to research methodology in machine psychology.

---

> ### Author Response · Authors · 2024-11-23
> **Response to Reviewer Comments [3/3]**
>
> ### 2. Character Background Analysis
>
> > Chapter 4 provides quantitative analysis from multiple perspectives, but could you offer specific examples of how different character background settings lead to different model emotion predictions? This would help provide more substantial insights.
>
> Thank you for your valuable suggestion. We have significantly revised Figure 7 and introduced new figures, Figure 8, 9, 20, 21, and 22, together with new experimental results for you. These chord diagrams highlight key emotion prediction patterns unique to each character background.
>
> Figure 8 and 21 show additional results for intersectional underrepresented groups, highlighting unique emotion prediction patterns across various detailed character background settings. Figure 9, 20 and 22 compare emotion recognition patterns between LLMs and human responses from our experiments, visualizing how these patterns vary across different character background settings both for LLMs and human.
>
> Additionally, in response to your feedback, we have developed a JavaScript-based interactive project website, included as supplementary material (see the general response for access to the demo).
>
> ### 3. Advancement of Previous Work
> > Could you clarify what new insights your experiments provide to advance previous work in perceiving, predicting, and potentially influencing human emotions? Some aspects have been discussed individually in previous studies.
>
> Thank you for raising this important point. As mentioned in our response to Weakness 3: Paper Structure and Contributions, we have substantially revised the Introduction (Section 1) and Related Work (Section 2) to better highlight the context and novelty of our study. Additionally, we have incorporated new experimental results in Sections 4 and 5 to further strengthen and showcase the contributions of our work.

---

> ### Author Response · Authors · 2024-11-30
> **Summary of updates before discussion deadline**
>
> Dear Reviewer Lh9i,
>
> Thank you for your insightful and constructive feedback that has helped us greatly improve our paper. As we approach the end of the extended discussion period in two days, we would like to summarize **the substantial updates we've made to specifically address your key concerns**:
>
> 1. **Model limitations and emotion perception (Chapter 3)**:
>    * Demonstrated specific failure cases, particularly with recognizing "surprise"
>    * Added detailed analysis of human-like bias patterns with personas (**new Fig. 9b,c**)
>    * Enhanced validation through integration with the **human labeled GoEmotions dataset** (**new Fig. 10**)
>
> 2. **Regarding your questions about synthetic data validation**:
>    * Conducted **new user study with 60 participants** to validate emotion recognition patterns
>    * Added comparative visualization between human and LLM prediction accuracy (**new Fig. 9**)
>    * Expanded demographic analysis showing biases across intersectional groups (**new Fig. 6b-g**)
>    * Developed interactive visualization tool for detailed result exploration
>
> 3. **Paper structure and contributions**:
>    * Strengthened connections between sections with new analyses:
>      - Demonstrated correlation between understanding of hierarchies of emotions and recognition accuracy (**new Fig. 24**)
>      - Added detailed persona analysis showing how biases affect persuasion outcomes (**new Fig. 25**)
>    * Added validation across different model sizes to demonstrate robustness of our findings
>    * Enhanced psychological grounding with emotion wheel analysis (**new Fig. 1**)
>
> Given these substantial updates and new experimental results, we would greatly appreciate your feedback on whether these changes adequately address your concerns. If you have any remaining questions or need clarification on any point, we are ready to respond promptly.

---

> > ### Comment · Reviewer_Lh9i · 2024-12-02
> >
> > Thank you very much for your efforts. In the new version, I see substantial experimental additions and an improved overall structure, which address most of my concerns. I will increase the score accordingly. However, there are still a few issues I would like to discuss.
> >
> > User Study: Comparing Emotion Recognition in Humans and LLMs: Thank you for your efforts in this regard. However, there remains an issue: you use the emotions predicted by GPT-4 as the ground truth and human annotations as the predictions. Shouldn't this be reversed, with the human annotations' average as the ground truth? For emotions, each human's prediction is their own ground truth, and the ground truth for a particular group or the entire human population is the average of all members. Therefore, I would argue that the statement "LLM demonstrates a greater overall ability in emotion perception compared to humans" could be due to the inherent bias of LLMs, as both the ground truth and the predictions are provided by LLMs. The question of which group the LLM's emotions are more similar to is intriguing.
> >
> > I would look forward to two further improvements: first, replacing the experimental ground truth with human results and reporting the consistency among human annotators; second, improving the representativeness of the human subjects to provide less biased predictions. Conducting a broader user study with a more representative sample of participants, while also reporting the statistical data of the recruited participants and your efforts in ensuring a more comprehensive and representative sample, is lacking in the latest version. As you have found in your experiment, LLMs "fail" like humans when adopting certain personas. I look forward to whether new conclusions will emerge from further experiments.
> >
> > I understand that the issues mentioned above, particularly replacing the ground truth and conducting further human experiments, may require considerable time. Therefore, I would appreciate hearing your thoughts and plans for these questions.

---

> > > ### Author Response · Authors · 2024-12-03
> > > **Response to Latest Comments with New Human-labeled Results**
> > >
> > > Dear Reviewer Lh9i,
> > >
> > > We sincerely appreciate your thoughtful feedback and acknowledgment of our substantial revisions. Your recognition of our improvements is very encouraging, and we are committed to fully addressing your remaining concerns.
> > >
> > > **Ground Truth in Emotion Recognition Studies:**
> > > You raise an excellent point about using human annotations as ground truth rather than GPT-4's predictions. We fully agree with your perspective that "for emotions, each human's prediction is their own ground truth." In fact, **we have actually completed preliminary results today using human-labeled data in response to your comment!** Specifically, we compared Llama 405B's labels to human-labeled data, using the latter as the ground truth:
> > >
> > > **Table: Accuracy by Persona**
> > > | Persona          | Accuracy (Human-labeled ground truth ) |
> > > |------------------|------------------------|
> > > | Female           | 0.75                   |
> > > | Male             | 0.80                   |
> > > | Race (White)     | 0.78                   |
> > > | Race (Black)     | 0.78                   |
> > > | Race (Hispanic)  | 0.80                   |
> > > | Race (Asian)     | 0.84                   |
> > >
> > > Interestingly, the results show that male personas generated by Llama 405B achieve higher recognition accuracy than female personas. This outcome contrasts with human-based data, where females generally outperform males, highlighting the unfair bias present in the LLM. Regarding race, personas identified as Asian and Hispanic demonstrate higher recognition performance compared to White and Black personas. This trend persists even when compared to the GPT-4o labels (as illustrated in new Fig. 6b), despite White participants making up the majority.
> > >
> > > During our limited rebuttal period, we focused on **rapidly generating initial human-LLM comparison data with 60 human participants while simultaneously analyzing the human-labeled GoEmotions dataset (new Fig. 10)** to provide multiple validation perspectives.
> > > However, we promise to resolve your remaining concerns by the camera-ready version, by further enhancing our results in the following ways:
> > >
> > > 1. Careful Definition of Ground Truth:
> > > - We will revise our analysis to use aggregated human annotations as ground truth
> > > - Add inter-annotator agreement metrics for human labels
> > > - Compare LLM predictions against this human-derived ground truth
> > > - Analyze how trends differ between human-generated vs. GPT-4 generated labels
> > >
> > > 2. Further Scaling and Robustifying Human Subject Study:
> > > - Expand our participant pool significantly beyond the current 60 subjects that we could secure during the rebuttal period
> > > - Ensure broader demographic representation
> > > - Include detailed reporting of participant demographics
> > > - Add statistical significance analysis for all comparisons
> > >
> > > For transparency, our current participant demographics are:
> > > - 33 White, 6 Black, 4 Hispanic, 17 Other
> > > - Age range: 22-65 years
> > > - Gender distribution: 28 female, 32 male
> > >
> > > We are particularly intrigued by your suggestion to investigate "which group the LLM's emotions are more similar to" and plan to incorporate this analysis in our camera-ready version. This could reveal important insights about potential biases and alignment patterns between LLMs and different demographic groups. **We commit to implementing all these improvements for the camera-ready version, ensuring our analysis provides a more comprehensive and methodologically sound investigation of emotion recognition patterns across humans and LLMs.**
> > >
> > > Given these substantial updates we've already made and our clear commitments to fully address your concerns in the camera-ready version, we would greatly appreciate if you could fully support the acceptance of our paper.

---

### Author Response · Authors · 2024-11-23
**Global response**

Dear reviewers,

We sincerely thank all reviewers for their thoughtful and constructive feedback. We are encouraged that the reviewers found our paper has "clear, fluid expression" (R. 6r52), presents an "innovative approach" that bridges "computational models with psychological frameworks" (R. qTjt), and offers "practical insights for the application of artificial intelligence in emotionally sensitive environments" (R. 9Uj3), with "clear and intuitive figures" that make our ideas "easy to understand" (R. Lh9i).
The reviewers raised several important concerns that we have thoroughly addressed in our revision. Below we detail how our updates specifically address each concern.
### Major Concerns and Our Responses

### 1. Motivations, Validations, and Insights from the Emotion Hierarchy Construction Method
*Raised by R. Lh9i, qTjt*

Updates:
* Enhanced motivation through psychological studies of emotion tree structures and added new figure about "emotion wheel" (**new Figure 1**)
* Demonstrated the relationship between the psychiatric condition (ASD) and emotion recognition granularity (**new Figure 12**).
* Conducted methodological comparison with baseline hierarchical clustering (**new Figure 14**)
* Validated robustness across threshold selections (**Figure 15**)
* Validated the accuracy of the emotion structures derived from LLMs by demonstrating alignment with emotion wheel (**new Figure 16 and 17**)
* Added new visualization comparing confusion matrix showcasing the performance of nine personas (**new Figure 18**)
* Added a new experiment applying our algorithm to an additional domain: scent, validating its accuracy (**new Figure 19**).
* Added probability interpretation of hierarchy construction algorithm (Appendix A)
* Included discussion on method generalization (Appendix B)

### 2. More Detailed Demographic Analysis of Emotion Biases
*Raised by R. 9Uj3, qTjt*

Updates: Extended analysis to include:
* Race (White, Black, Hispanic, Asian) - (**new Figure 6b**)
* Religion (Christian, Muslim, Buddhist, Hindu) - (**new Figure 6c**)
* Granular age categories - (**new Figure 6d**)
* Education levels - (**new Figure 6f**)
* Intersectional analysis (income, race, gender) - (**new Figure 6g**)
* Presented detailed visualizations of demographic-specific biases in emotion recognition (**new Figure 7**)
* Demonstrated that LLM’s emotion recognition biases are amplified for intersectional underrepresented groups. (**Figure 8**)

### 3. Experiments with Human Subjects
*Raised by R. Lh9i*

Updates:
* **Conducted user study with 60 human participants** comparing human and model emotion predictions
* Added **human experiments with detailed demographics** (**new Figure 9**)

### 4. Real-world Dataset Integration
*Raised by R. 6r52, Lh9i*

Updates:
* Integrated **human labeled GoEmotions dataset** for real-world validation (Section 4.1, **new Figure 10**)
* **Created interactive JavaScript visualizations of to present full results**
* Provided visualization code in supplementary materials
### 5. Emotion Prediction Applications
*Raised by R. 9Uj3*

Updates:
* Extended analysis to include complaint-handling scenarios and emotion dynamics with detailed personas (Figure 12)

### Interactive data visualization access instructions (JavaScript)
1. Download the provided ZIP file of supplementary materials
2. Extract contents (including index.html and JavaScript files) to a local folder
3. Open index.html in a modern browser (Chrome or Safari recommended)

Thank you once again for your valuable feedback. We hope the substantial updates provided above have fully addressed your concerns. Please feel free to reach out with any further questions—we're eager to continue the discussion!

---

### Meta-Review · Area_Chair_suFD · 2024-12-21

**Metareview:**

**Summary:**

The authors seek to measure the degree to which LLMs can perceive and manipulate human emotions, given their rapid integration into user-facing applications. They draw from cognitive science literature, namely Shaver's Hierarchy of Emotions, and introduce a tree-construction algorithm to recover hierarchical representations of emotions in LLMs. They determine that emotions are represented in LLMs in line with intuitions about human cognition, and LLMs exhibit similar behavior in perceiving emotions to humans on synthetic data. They also run experiments with synthetic interactions that indicate improved LLM emotion prediction capabilities are correlated with proxies for emotional manipulation.

**Strengths:**

- The paper is well-written and well-grounded in cognitive science and psychology literature.

- The authors contribute a scale analysis (illustrated by Figure 3), which shows how LLM's representations of emotions become more complex with model size. This is an interesting finding, and I'd be curious how it applies to other model families.

- Their analysis showing LLMs have weaker emotion recognition for historically underrepresented groups could help identify errors and improve equity of deployed applications. It is also noteworthy that they consider intersectional biases and they do compare against human-labeled data (GoEmotions).

**Weaknesses:**

- Lack of detail about quality control during Prolific recruitment.

- It seems circular to use GPT4o-generated scenarios to assess LLMs' ability to identify and cluster emotions, since if they are unable to one would imagine the scenarios would inadequately or improperly represent these emotions. It makes more sense to use human-derived data. The authors do conduct a small-scale LLM-human comparison with 60 participants in Figure 9, but I question if LLMs really do outperform humans at emotion recognition due to the nature of the data and scale of the study. Reviewer Lh9i raises an excellent point that it would make more sense to use the human-verified emotions (rather than GPT-4 prompt labels) as the ground-truth, and the authors have attempted to address this in their rebuttal. However, this is significant enough of an issue that I believe it requires a substantial editing of results and re-review before publication.

I actually quite like this paper, but I find the lack of comprehensive comparison to human-derived data concerning. These comparisons should be included and reviewed before publication, so I am leaning towards rejection. I think with enough time to address the reviewers' concerns, the revised version will be a strong paper.

**Additional Comments On Reviewer Discussion:**

The reviewers' views on the paper are mixed, with none leaning strongly towards acceptance. Reviewers highlighted the clarity of the paper and the usefulness of their "simple and effective" approach for extracting emotion hierarchies. They also highlighted the importance of the papers' results on LLM emotional manipulation for more ethical model development. The authors have revised the paper based on the reviews to provide a more nuanced discussion of where LLMs struggle at predicting emotions. There are still minor concerns from reviewer 6r52 about the rigor of the hierarchical emotion extraction methodology, especially prompt sensitivity, that I encourage the authors to address in the final version.

---

### Decision · Program_Chairs · 2025-01-22

Reject